# Firefly Neural Architecture Descent: a General Approach for Growing Neural Networks

**Lemeng Wu**[*]
Department of Computer Science
University of Texas at Austin
Austin, TX 78712
`lmwu@cs.utexas.edu`

**Bo Liu**[*]
Department of Computer Science
University of Texas at Austin
Austin, TX 78712
`bliu@cs.utexas.edu`

**Peter Stone**
Department of Computer Science
University of Texas at Austin
Austin, TX 78712
`pstone@cs.utexas.edu`

**Qiang Liu**
Department of Computer Science
University of Texas at Austin
Austin, TX 78712
`lqiang@cs.utexas.edu`

## Abstract

We propose *firefly neural architecture descent*, a general framework for progressively and dynamically growing neural networks to jointly optimize the networks' parameters and architectures. Our method works in a steepest descent fashion, which iteratively finds the best network within a functional neighborhood of the original network that includes a diverse set of candidate network structures. By using Taylor approximation, the optimal network structure in the neighborhood can be found with a greedy selection procedure. We show that firefly descent can flexibly grow networks both wider and deeper, and can be applied to learn accurate but resource-efficient neural architectures that avoid catastrophic forgetting in continual learning. Empirically, firefly descent achieves promising results on both neural architecture search and continual learning. In particular, on a challenging continual image classification task, it learns networks that are smaller in size but have higher average accuracy than those learned by the state-of-the-art methods.

## 1 Introduction

Although biological brains are developed and shaped by complex progressive growing processes, most existing artificial deep neural networks are trained under fixed network structures (or architectures). Efficient techniques that can progressively grow neural network structures can allow us to jointly optimize the network parameters and structures to achieve higher accuracy and computational efficiency, especially in dynamically changing environments. For instance, it has been shown that accurate and energy-efficient neural network can be learned by progressively growing the network architecture starting from a relatively small network (Liu et al., 2019; Wang et al., 2019). Moreover, previous works also indicate that knowledge acquired from previous tasks can be transferred to new and more complex tasks by expanding the network trained on previous tasks to a functionally-equivalent larger network to initialize the new tasks (Chen et al., 2016; Wei et al., 2016).

In addition, dynamically growing neural network has also been proposed as a promising approach for preventing the challenging *catastrophic forgetting* problem in continual learning (Rusu et al., 2016; Yoon et al., 2017; Rosenfeld & Tsotsos, 2018; Li et al., 2019).

---

[*]Equal contribution.

Unfortunately, searching for the optimal way to grow a network leads to a challenging combinatorial optimization problem. Most existing works use simple heuristics (Chen et al., 2016; Wei et al., 2016), or random search (Elsken et al., 2017, 2018) to grow networks and may not fully unlock the power of network growing. An exception is splitting steepest descent (Liu et al., 2019), which considers growing networks by splitting the existing neurons into multiple copies, and derives a principled functional steepest-descent approach for determining which neurons to split and how to split them. However, the method is restricted to neuron splitting, and can not incorporate more flexible ways for growing networks, including adding brand new neurons and introducing new layers.

In this work, we propose *firefly neural architecture descent*, a general and flexible framework for progressively growing neural networks. Our method is a local descent algorithm inspired by the typical gradient descent and splitting steepest descent. It grows a network by finding the best larger networks in a *functional neighborhood* of the original network whose size is controlled by a step size $\epsilon$, which contains a rich set of networks that have various (more complex) structures, but are $\epsilon$-close to the original network in terms of the function that they represent. The key idea is that, when $\epsilon$ is small, the combinatorial optimization on the functional neighborhood can be simplified to a greedy selection, and therefore can be solved efficiently in practice.

The firefly neural architecture descent framework is highly flexible and practical and allows us to derive general approaches for growing wider and deeper networks (Section 2.2-2.3). It can be easily customized to address specific problems. For example, our method provides a powerful approach for dynamic network growing in continual learning (Section 2.4), and can be applied to optimize cell structures in cell-based neural architecture search (NAS) such as DARTS (Liu et al., 2018b) (Section 3). Experiments show that Firefly efficiently learns accurate and resource-efficient networks in various settings. In particular, for continual learning, our method learns more accurate and smaller networks that can better prevent catastrophic forgetting, outperforming state-of-the-art methods such as Learn-to-Grow (Li et al., 2019) and Compact-Pick-Grow (Hung et al., 2019a).

## 2 Firefly Neural Architecture Descent

In this section, we start with introducing the general framework (Section 2.1) of firefly neural architecture descent. Then we discuss how the framework can be applied to grow a network both wider and deeper (Section 2.2-2.3). To illustrate the flexibility of the framework, we demonstrate how it can help tackle catastrophic forgetting in continual learning (Section 2.4).

### 2.1 The General Framework

We start with the general problem of jointly optimizing neural network parameters and model structures. Let $\Omega$ be a space of neural networks with different parameters and structures (e.g., networks of various widths and depths). Our goal is to solve

$$\arg\min_f \left\{ L(f) \quad s.t. \quad f \in \Omega, \quad C(f) \leq \eta \right\}, \tag{1}$$

where $L(f)$ is the training loss function and $C(f)$ is a complexity measure of the network structure that reflects the computational or memory cost. This formulation poses a highly challenging optimization problem in a complex, hierarchically structured space.

We approach (1) with a steepest descent type algorithm that generalizes typical parametric gradient descent and the splitting steepest descent of Liu et al. (2019), with an iterative update of the form

$$f_{t+1} = \arg\min_f \left\{ L(f) \quad s.t. \quad f \in \partial(f_t, \epsilon), \quad C(f) \leq C(f_t) + \eta_t \right\}, \tag{2}$$

where we find the best network $f_{t+1}$ in neighborhood set $\partial(f_t, \epsilon)$ of the current network $f_t$ in $\Omega$, whose complexity cannot exceed that of $f_t$ by more than a threshold $\eta_t$. Here $\partial(f_t, \epsilon)$ denotes a neighborhood of $f_t$ of "radius" $\epsilon$ such that $f(x) = f_t(x) + O(\epsilon)$ for $\forall f \in \partial(f_t, \epsilon)$. $\epsilon$ can be viewed as a small step size, which ensures that the network changes smoothly across iterations, and importantly, allows us to use Taylor expansion to significantly simplify the optimization (2) to yield practically efficient algorithms.

The update rule in (2) is highly flexible and reduces to different algorithms with different choices of $\eta_t$ and $\partial(f_t, \epsilon)$. In particular, when $\epsilon$ is infinitesimal, by taking $\eta_t = 0$ and $\partial(f_t, \epsilon)$ the typical

---
**Algorithm 1** Firefly Neural Architecture Descent
---
**Input**: Loss function $L(f)$; initial small network $f_0$; search neighborhood $\partial(f, \epsilon)$; maximum increase of size $\{\eta_t\}$.

**Repeat:** At the $t$-th growing phase:

**1.** Optimize the parameter of $f_t$ with fixed structure using a typical optimizer for several epochs.

**2.** Minimize $L(f)$ in $f \in \partial(f, \epsilon)$ without the complexity constraint (see e.g., (4)) to get a large "over-grown" network $\tilde{f}_{t+1}$ by performing gradient descent.

**3.** Select the top $\eta_t$ neurons in $\tilde{f}_{t+1}$ with the highest importance measures to get $f_{t+1}$ (see (5)).

---

Euclidean ball on the parameters, (2) reduces to standard gradient descent which updates the network parameters with architecture fixed. However, by taking $\eta_t > 0$ and $\partial(f_t, \epsilon)$ a rich set of neural networks with different, larger network structures than $f_t$, we obtain novel *architecture descent* rules that allow us to incrementally grow networks.

In practice, we alternate between parametric descent and architecture descent according to a user-defined schedule (see Algorithm 2.1). Because architecture descent increases the network size, it is called less frequently (e.g., only when a parametric local optimum is reached). From the optimization perspective, performing architecture descent allows us to lift the optimization into a higher dimensional space with more parameters, and hence escape local optima that cannot be escaped in the lower dimensional space (of the smaller models).

In the sequel, we instantiate the neighborhood $\partial(f_t, \epsilon)$ for growing wider and deeper networks, and for continual learning, and discuss how to solve the optimization in (2) efficiently in practice.

## 2.2 Growing Network Width

We discuss how to define $\partial(f_t, \epsilon)$ to progressively build increasingly wider networks, and then introduce how to efficiently solve the optimization in practice. We illustrate the idea with two-layer networks, but extension to multiple layers works straightforwardly. Assume $f_t$ is a two-layer neural network (with one hidden layer) of the form $f_t(x) = \sum_{i=1}^{m} \sigma(x, \theta_i)$, where $\sigma(x, \theta_i)$ denotes its $i$-th neuron with parameter $\theta_i$ and $m$ is the number of neurons (a.k.a. width). There are two ways to introduce new neurons to build a wider network, including splitting existing neurons in $f_t$ and introducing brand new neurons; see Figure 1.

**Splitting Existing Neurons**   Following Liu et al. (2019), an essential approach to growing neural networks is to split the neurons into a linear combination of multiple similar neurons. Formally, splitting a neuron $\theta_i$[2] into a set of neurons $\{\theta_{i\ell}\}$ with weights $\{w_{i\ell}\}$ amounts to replacing $\sigma(x, \theta_i)$ in $f_t$ with $\sum_{\ell} w_{i\ell}\sigma(x, \theta_{i\ell})$. We shall require that $\sum_{\ell} w_{i\ell} = 1$ and $\|\theta_{i\ell} - \theta\|_2 \leq \epsilon$, $\forall \ell$ so that the new network is $\epsilon$-close to the original network. As argued in Liu et al. (2019), when $f_t$ reaches a parametric local optimum and $w_{i\ell} \geq 0$, it is sufficient to consider a simple binary splitting scheme, which splits a neuron $\theta_i$ into two equally weighted copies along opposite update directions, that is, $\sigma(x, \theta_i) \Rightarrow \frac{1}{2}\big(\sigma(x, \theta_i + \epsilon\delta_i) + \sigma(x, \theta_i - \epsilon\delta_i)\big)$, where $\delta_i$ denotes the update direction.

**Growing New Neurons**   Splitting the existing neurons yields a "local" change because the parameters of the new neurons are close to that of the original neurons. A way to introduce "non-local" updates is to add brand new neurons with arbitrary parameters far away from the existing neurons. This is achieved by replacing $f_t$ with $f_t(x) + \epsilon\sigma(x, \delta)$, where $\delta$ now denotes a trainable parameter of the new neuron and the neuron is multiplied by $\epsilon$ to ensure the new network is close to $f_t$ in function.

Overall, to grow $f_t(x) = \sum_i \sigma(x; \theta_i)$ wider, the neighborhood set $\partial(f_t, \epsilon)$ can include functions of the form

$$f_{\boldsymbol{\varepsilon}, \boldsymbol{\delta}}(x) = \sum_{i=1}^{m} \frac{1}{2}\Big(\sigma(x, \theta_i + \varepsilon_i\delta_i) + \sigma(x, \theta_i - \varepsilon_i\delta_i)\Big) + \sum_{i=m+1}^{m+m'} \varepsilon_i\sigma(x, \delta_i),$$

where we can potentially split all the neurons in $f_t$ and add upto $m'$ new non-local neurons ($m'$ is a hyperparameter). Whether each new neuron will eventually be added is controlled by an

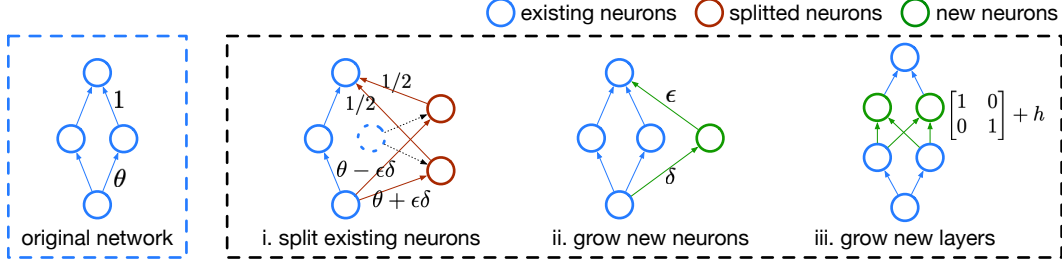

Figure 1: An illustration of three different growing methods within firefly neural architecture descent. Both $\delta$ and $h$ are trainable perturbations.

individual step-size $\varepsilon_i$ that satisfies $|\varepsilon_i| \leq \epsilon$. If $\varepsilon_i = 0$, it means the corresponding new neuron is not introduced. Therefore, the number of new neurons introduced in $f_{\varepsilon,\delta}$ equals the $\ell_0$ norm $\|\varepsilon\|_0 := \sum_{i=1}^{m+m'} \mathbb{I}(\varepsilon_i = 0)$. Here $\varepsilon = [\varepsilon_i]_{i=1}^{m+m'}$ and $\delta = [\delta_i]_{i=1}^{m+m'}$.

Under this setting, the optimization in (2) can be framed as

$$\min_{\varepsilon,\delta} \left\{ L(f_{\varepsilon,\delta}) \quad s.t. \quad \|\varepsilon\|_0 \leq \eta_t, \quad \|\varepsilon\|_\infty \leq \epsilon, \quad \|\delta\|_{2,\infty} \leq 1 \right\}, \tag{3}$$

where $\|\delta\|_{2,\infty} = \max_i \|\delta_i\|_2$, which is constructed to prevent $\|\delta_i\|_2$ from becoming arbitrarily large.

**Optimization** It remains to solve the optimization in (3), which is challenging due to the $\ell_0$ constraint on $\varepsilon$. However, when the step size $\epsilon$ is small, we can solve it approximately with a simple two-step method: we first optimize $\delta$ and $\varepsilon$ while dropping the $\ell_0$ constraint, and then re-optimize $\varepsilon$ with Taylor approximation on the loss, which amounts to simply picking the new neurons with the largest contribution to the decrease of loss, measured by the gradient magnitude.

*Step One.* Optimizing $\delta$ and $\varepsilon$ without the sparsity constraint $\|\varepsilon\|_0 \leq \eta_t$, that is,

$$[\tilde{\varepsilon}, \tilde{\delta}] = \arg\min_{\varepsilon,\delta} \left\{ L(f_{\varepsilon,\delta}) \quad s.t. \quad \|\varepsilon\|_\infty \leq \epsilon, \quad \|\delta\|_{2,\infty} \leq 1 \right\}. \tag{4}$$

In practice, we solve the optimization with gradient descent by turning the constraint into a penalty. Because $\epsilon$ is small, we only need to perform a small number of gradient descent steps.

*Step Two.* Re-optimizing $\varepsilon$ with Taylor approximation on the loss. To do so, note that when $\epsilon$ is small, we have by Taylor expansion:

$$L(f_{\varepsilon,\tilde{\delta}}) = L(f) + \sum_{i=1}^{m+m'} \varepsilon_i s_i + O(\epsilon^2), \qquad s_i = \frac{1}{\tilde{\varepsilon}_i} \int_0^{\tilde{\varepsilon}_i} \nabla_{\zeta_i} L(f_{[\tilde{\varepsilon}_{\neg i}, \zeta_i], \tilde{\delta}}) d\zeta_i,$$

where $[\tilde{\varepsilon}_{\neg i}, \zeta_i]$ denotes replacing the $i$-th element of $\tilde{\varepsilon}$ with $\zeta_i$, and $s_i$ is an integrated gradient that measures the contribution of turning on the $i$-th new neuron. In practice, we approximate the integration in $s_i$ by discrete sampling: $s_i \approx \frac{1}{n} \sum_{z=1}^n \nabla_{c_z} L(f_{[\tilde{\varepsilon}_{\neg i}, c_z], \tilde{\delta}})$ with $c_z = (2z-1)/2n\tilde{\varepsilon}_i$ and $n$ a small integer (e.g., 3). Therefore, optimizing $\varepsilon$ with fixed $\delta = \tilde{\delta}$ can be approximated by

$$\hat{\varepsilon} = \arg\min_{\varepsilon} \left\{ \sum_{i=1}^{m+m'} \varepsilon_i s_i \quad s.t. \quad \|\varepsilon\|_0 \leq \eta_t, \quad \|\varepsilon\|_\infty \leq \epsilon \right\}. \tag{5}$$

It is easy to see that finding the optimal solution reduces to selecting the neurons with the largest gradient magnitude $|s_i|$. Precisely, we have $\hat{\varepsilon}_i = -\epsilon \, \mathbb{I}(|s_i| \geq |s_{(\eta_t)}|) \, \text{sign}(s_i)$, where $|s_{(1)}| \leq |s_{(2)}| \leq \cdots$ is the increasing ordering of $\{|s_i|\}$. Finally, we take $f_{t+1} = f_{\hat{\varepsilon},\tilde{\delta}}$.

It is possible to further re-optimize $\delta$ with fixed $\varepsilon$ and repeat the alternating optimization iteratively. However, performing the two steps above is computationally efficient and already solves the problem reasonably well as we observe in practice.

**Remark** When we include only neural splitting in $\partial(f_t, \epsilon)$, our method is equivalent to splitting steepest descent (Liu et al., 2019), but with a simpler and more direct gradient-based optimization rather than solving the eigen-problem in Liu et al. (2019); Wang et al. (2019).

## 2.3  Growing New Layers

We now introduce how to grow new layers under our framework. The idea is to include in $\partial(f_t, \epsilon)$ deeper networks with extra trainable residual layers and to select the layers (and their neurons) that contribute the most to decreasing the loss using the similar two-step method described in Section 2.2.

Assume $f_t$ is a $d$-layer deep neural network of form $f_t = g_d \circ \cdots \circ g_1$, where $\circ$ denotes function composition. In order to grow new layers, we include in $\partial(f_t, \epsilon)$ functions of the form

$$f_{\boldsymbol{\varepsilon}, \boldsymbol{\delta}} = g_d \circ (I + h_{d-1}) \cdots (I + h_2) \circ g_2 \circ (I + h_1) \circ g_1, \quad \text{with} \quad h_\ell(\cdot) = \sum_{i=1}^{m'} \varepsilon_{\ell i} \sigma(\cdot, \delta_{\ell i}),$$

in which we insert new residual layers of form $I + h_\ell$; here $I$ is the identity map, and $h_\ell$ is a layer that can consist of upto $m'$ newly introduced neurons. Each neuron in $h_\ell$ is associated with a trainable parameter $\delta_{\ell i}$ and multiplied by $\varepsilon_{\ell i} \in [-\epsilon, \epsilon]$. As before, the $(\ell i)$-th neuron is turned off if $\varepsilon_{\ell i} = 0$, and the whole layer $h_\ell$ is turned off if $\varepsilon_{\ell i} = 0$ for all $i \in [1, m']$. Therefore, the number of new neurons introduced in $f_{\boldsymbol{\varepsilon}, \boldsymbol{\delta}}$ equals $\|\boldsymbol{\varepsilon}\|_0 := \sum_{i\ell} \mathbb{I}(\epsilon_{i\ell} \neq 0)$, and the number of new layers added equals $\|\boldsymbol{\varepsilon}\|_{\infty, 0} := \sum_\ell \mathbb{I}(\max_i |\varepsilon_{\ell i}| \neq 0)$. Because adding new neurons and new layers have different costs, they can be controlled by two separate budget constraints (denoted by $\eta_{\eta_{t,0}}$ and $\eta_{t,1}$, respectively). Then the optimization of the new network can be framed as

$$\min_{\boldsymbol{\varepsilon}, \boldsymbol{\delta}} \Big\{ L(f_{\boldsymbol{\varepsilon}, \boldsymbol{\delta}}) \quad s.t. \quad \|\boldsymbol{\varepsilon}\|_0 \leq \eta_{t,0}, \ \ \|\boldsymbol{\varepsilon}\|_{\infty, 0} \leq \eta_{t,1}, \ \ \|\boldsymbol{\varepsilon}\|_\infty \leq \epsilon, \ \ \|\boldsymbol{\delta}\|_{2,\infty} \leq 1 \Big\},$$

where $\|\boldsymbol{\delta}\|_{2,\infty} = \max_{\ell, i} \|\delta_{\ell i}\|_2$. This optimization can be solved with a similar two-step method to the one for growing width, as described in Section 2.2: we first find the optimal $[\tilde{\epsilon}, \tilde{\boldsymbol{\delta}}]$ without the complexity constraints (including $\|\boldsymbol{\varepsilon}\|_0 \leq \eta_{t,0}$, $\|\boldsymbol{\varepsilon}\|_{0,\infty} \leq \eta_{t,1}$), and then re-optimize $\boldsymbol{\varepsilon}$ with a Taylor approximation of the objective:

$$\min_{\boldsymbol{\varepsilon}} \left\{ \sum_{\ell i} \epsilon_{\ell i} s_{\ell i} \quad s.t. \quad \|\boldsymbol{\varepsilon}\|_0 \leq \eta_{t,0}, \ \ \|\boldsymbol{\varepsilon}\|_{\infty, 0} \leq \eta_{t,1} \right\}, \text{ where } s_{\ell i} = \frac{1}{\tilde{\varepsilon}_{\ell i}} \int_0^{\tilde{\varepsilon}_{\ell i}} \nabla_{\zeta_{\ell i}} L(f_{[\tilde{\boldsymbol{\varepsilon}}_{\neg \ell i}, \zeta_{\ell i}], \tilde{\boldsymbol{\delta}}}) d\zeta_{\ell i}.$$

The solution can be obtained by sorting $|s_{ti}|$ in descending order and select the top-ranked neurons until the complexity constraint is violated.

**Remark**  In practice, we can apply all methods above to simultaneously grow the network wider and deeper. Firefly descent can also be extended to various other growing settings without case-by-case mathematical derivation. Moreover, the space complexity to store all the intermediate variables is $\mathcal{O}(N + m')$, where $N$ is the size of the sub-network we consider expanding and $m'$ is the number of new neuron candidates.[3]

## 2.4  Growing Networks in Continual Learning

Continual learning (CL) studies the problem of learning a sequence of different tasks (datasets) that arrive in a temporal order, so that whenever the agent is presented with a new task, it no longer has access to the previous tasks. As a result, one major difficulty of CL is to avoid *catastrophic forgetting*, in that learning the new tasks severely interferes with the knowledge learned previously and causes the agent to "forget" how to do previous tasks. One branch of approaches in CL consider dynamically growing networks to avoid *catastrophic forgetting* (Rusu et al., 2016; Li & Hoiem, 2017; Yoon et al., 2017; Li et al., 2019; Hung et al., 2019a). However, most existing growing-based CL methods use hand-crafted rules to expand the networks (e.g. uniformly expanding each layer) and do not explicitly seek for the best growing approach under a principled optimization framework. We address this challenge with the Firefly architecture descent framework.

Let $\mathbb{D}_t$ be the dataset appearing at time $t$ and $f_t$ be the network trained for $\mathbb{D}_t$. At each step $t$, we maintain a master network $f_{1:t}$ consisting of the union of all the previous networks $\{f_s\}_{s=1}^t$, such that each $f_s$ can be retrieved by applying a proper binary mask. When a new task $\mathbb{D}_{t+1}$ arrives, we construct $f_{t+1}$ by leveraging the existing neurons in $f_{1:t}$ as much as possible, while adding a controlled number of new neurons to capture the new information in $\mathbb{D}_{t+1}$.

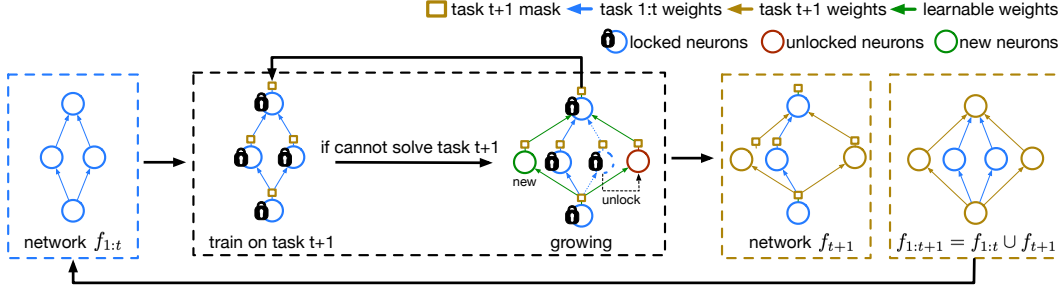

Figure 2: Illustration of how Firefly grows networks in continual learning.

Specifically, we design $f_{t+1}$ to include three types of neurons (see Figure 2): **1)** *Old neurons from* $f_{1:t}$, whose parameters are *locked* during the training of $f_{t+1}$ on the new task $\mathbb{D}_{t+1}$. This does not introduce extra memory cost. **2)** *Old neurons from* $f_t$, whose parameters are *unlocked and updated* during the training of $f_{t+1}$ on $\mathbb{D}_{t+1}$. This introduces new neurons and hence increases the memory size. It is similar to network splitting in Section 2.2 in that the new neurons are evolved from an old neuron, but only one copy is generated and the original neuron is not discarded. **3)** *New neurons* introduced in the same way as in Section 2.2[4] which also increases the memory cost. Overall, assuming $f_{1:t}(x) = \sum_{i=1}^{m} \sigma(x; \theta_i)$, possible candidates of $f_{t+1}$ indexed by $\boldsymbol{\varepsilon}, \boldsymbol{\delta}$ are of the form:

$$f_{\boldsymbol{\varepsilon},\boldsymbol{\delta}}(x) = \sum_{i=1}^{m} \sigma(x; \theta_i + \varepsilon_i \delta_i) + \sum_{i=m+1}^{m+m'} \varepsilon_i \sigma(x; \delta_i),$$

where $\varepsilon_i \in [-\epsilon, \epsilon]$ again controls if the corresponding neuron is locked or unlocked (for $i \in [m]$), or if the new neuron should be introduced (for $i > m$). The new neurons introduced into the memory are $\|\boldsymbol{\varepsilon}\|_0 = \sum_{i=1}^{m+m'} \mathbb{I}(\varepsilon \neq 0)$. The optimization of $f_{t+1}$ can be framed as

$$f_{t+1} = \arg\min_{\boldsymbol{\varepsilon},\boldsymbol{\delta}} \left\{ L(f_{\boldsymbol{\varepsilon},\boldsymbol{\delta}}; \mathbb{D}_{t+1}) \quad s.t. \quad \|\boldsymbol{\varepsilon}\|_0 \leq \eta_t, \ \|\boldsymbol{\varepsilon}\|_\infty \leq \epsilon, \ \|\boldsymbol{\delta}\|_{2,\infty} \leq 1 \right\},$$

where $L(f; \mathcal{D}_{t+1})$ denotes the training loss on dataset $\mathbb{D}_{t+1}$. The same two-step method in Section 2.2 can be applied to solve the optimization. After $f_{t+1}$ is constructed, the new master network $f_{1:t+1}$ is constructed by merging $f_{1:t}$ and $f_{t+1}$ and the binary masks of the previous tasks are updated accordingly. See Appendix A for the detailed algorithm.

## 3 Empirical Results

We conduct four sets of experiments to verify the effectiveness of firefly neural architecture descent. In particular, we first demonstrate the importance of introducing additional growing operations beyond neuron splitting (Liu et al., 2019) and then apply the firefly descent to both neural architecture search and continual learning problems. In both applications, firefly descent finds competitive but more compact networks in a relatively shorter time compared to state-of-the-art approaches.

**Toy RBF Network**   We start with growing a toy single-layer network to demonstrate the importance of introducing brand new neurons over pure neuron splitting. In addition, we show the local greedy selection in firefly descent is efficient by comparing it against random search. Specifically, we adopt a simple two-layer radial-basis function (RBF) network with one-dimensional input and compare various methods that grow the network gradually from 1 to 10 neurons. The training data consists of 1000 data points from a randomly generated RBF network. We consider the following methods: `Firefly`: firefly descent for growing wider by splitting neuron and adding upto $m' = 5$ brand new neurons; `Firefly (split)`: firefly descent for growing wider with only neuron splitting (e.g., $m' = 0$); `Splitting`: the steepest splitting descent of Liu et al. (2019); `RandSearch (split)`: randomly selecting one neuron and splitting in a random direction, repeated $k$ times to pick the best as the actual split; we take $k = 3$ to match the time cost with our method;

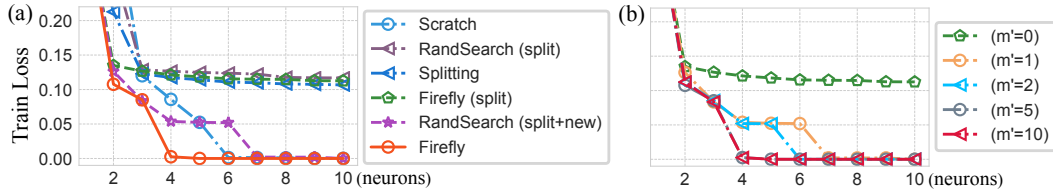

Figure 3: (a) Average training loss of different growing methods versus the number of grown neurons. (b) Firefly descent with different numbers of new neuron candidates.

`RandSearch (split+new)`: the same as `RandSearch (split)` but with 5 randomly initialized brand new neurons in the candidate during the random selecting; `Scratch`: training networks with fixed structures starting from scratch. We repeat each experiment 20 times with different ground-truth RBF networks and report the mean training loss in Figure 3(a).

As shown in Figure 3 (a), the methods with pure neuron splitting (without adding brand new neurons) can easily get stuck at a relatively large training loss and splitting further does not help escape the local minimum. In comparison, all methods that introduce additional brand new neurons can optimize the training loss to zero. Moreover, `Firefly` grows neural network the better than random search under the same candidate set of growing operations.

We also conduct a parameter sensitivity analysis on $m'$ in Figure 3(b), which shows the result of `Firefly` as we change the number $m'$ of the brand new neurons. We can see that the performance improves significantly by even just adding one brand new neuron in this case, and the improvement saturates when $m'$ is sufficiently large ($m' = 5$ in this case).

**Growing Wider and Deeper Networks**    We test the effectiveness of firefly descent for both growing network width and depth. We use VGG-19 (Simonyan & Zisserman, 2014) as the backbone network structure and compare our method with splitting steepest descent (Liu et al., 2019), Net2Net (Chen et al., 2016) which grows networks uniformly by randomly selecting the existing neurons in each layer, and neural architecture search by hill-climbing (NASH) (Elsken et al., 2017), which is a random sampling search method using network morphism on CIFAR-10. For Net2Net, the network is initialized as a thinner version of VGG-19, whose layers are $0.125\times$ the original sizes. For splitting steepest descent, NASH, and our method, we initialize the VGG-19 with 16 channels in each layer. For firefly descent, we grow a network by both splitting existing neurons and adding brand new neurons for widening the network; we add $m' = 50$ brand new neurons and set the budget to grow the size by $30\%$ at each step of our method. See Appendix B.2 for more information on the setting.

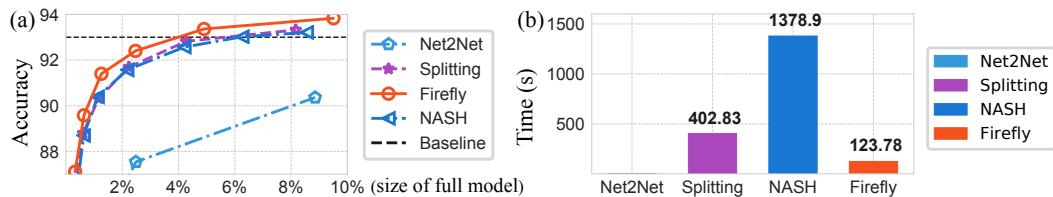

Figure 4: (a) Results of growing increasingly wider networks on CIFAR-10; VGG-19 is used as the backbone. (b) Computation time spent on growing for different methods.

Figure 4 (a) shows the test accuracy, where the x-axis is the percentage of the grown model's size over the standard VGG-19. We can see that the proposed method clearly outperforms the *splittting steepest descent* and Net2Net. In particular, we achieve comparable test accuracy as the full model with only $4\%$ of the full model's size. Figure 4(b) shows the average time cost of each growing method for one step, we can see that `Firefly` performs much faster than splitting the steepest descent and NASH. We also applied our method to gradually grow new layers in neural networks, we compare our method with NASH (Elsken et al., 2017) and AutoGrow (Wen et al., 2019). Due to the page limit, we defer the detailed results to Appendix B.2.

**Cell-Based Neural Architecture Search**    Next, we apply our method as a new way for improving cell-based Neural Architecture Search (NAS) (e.g. Zoph et al., 2018; Liu et al., 2018a; Real et al.,

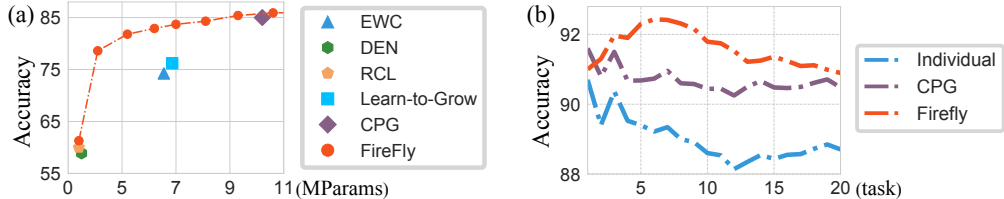

Figure 5: (a) Average accuracy on 10-way split of CIFAR-100 under different model size. We compare against Elastic Weight Consolidation (EWC) (Kirkpatrick et al., 2017), Dynamic Expandable Network (DEN) (Yoon et al., 2017), Reinforced Continual Learning (RCL) (Xu & Zhu, 2018) and Compact-Pick-Grow (CPG) (Hung et al., 2019a). (b) Average accuracy on 20-way split of CIFAR-100 dataset over 3 runs. Individual means train each task from scratch using the Full VGG-16.

2019). The idea of cell-based NAS is to learn optimal neural network modules (called cells), from a predefined search space, such that they serve as good building blocks to composite complex neural networks. Previous works mainly focus on using reinforcement learning or gradient based methods to learn a sparse cell structure from a predefined parametric template. Our method instead gradually grows a small parametric template during training and obtains the final network structure according to the growing pattern.

Following the setting in DARTS (Liu et al., 2018b), we build up the cells as computational graphs whose structure is the directed DAG with 7 nodes. The edges between the nodes are linear combinations of different computational operations (SepConv and DilConv of different sizes) and the identity map. To grow the cells, we apply firefly descent to grow the number of channels in each operation by both splitting existing neurons and adding brand new neurons. During search, we compose a network by stacking 5 cells sequentially to evaluate the quality of the cell structures. We train 100 epochs in total for searching, and grow the cells every 10 epochs. After training, the operation with the largest number of channels on edge is selected into the final cell structure. In addition, if the operations on the same edge all only grow a small amount of channels compared with the initial setting, we select the Identity operation instead. The network that we use in the final evaluation is a larger network consisting of 20 sequentially stacked cells. More details of the experimental setup can be found in Appendix B.3.

Table 1 reports the results comparing `Firefly` with several NAS baselines. Our method achieves a similar or better performance comparing with those RL-based and gradient-based methods like ENAS or DARTS, but with higher computational efficiency in terms of the total search time.

| Method | Search Time (GPU Days) | Param (M) | Error |
|---|---|---|---|
| NASNet-A (Zoph et al., 2018) | 2000 | 3.1 | 2.83 |
| ENAS (Pham et al., 2018) | 4 | 4.2 | 2.91 |
| Random Search | 4 | 3.2 | $3.29 \pm 0.15$ |
| DARTS (first order) (Liu et al., 2018b) | 1.5 | 3.3 | $3.00 \pm 0.14$ |
| DARTS (second order) (Liu et al., 2018b) | 4 | 3.3 | $2.76 \pm 0.09$ |
| Firefly | 1.5 | 3.3 | $2.78 \pm 0.05$ |

Table 1: Performance compared with several NAS baseline

**Continual Learning** Finally, we apply our method to grow networks for continual learning (CL), and compare with two state-of-the-art methods, Compact-Pick-Grow (CPG) (Hung et al., 2019a) and Learn-to-grow (Li et al., 2019), both of which also progressively grow neural networks for learning new tasks. For our method, we grow the networks starting from a thin variant of the original VGG-16 without fully connected layers.

Following the setting in Learn-to-Grow, we construct 10 tasks by randomly partitioning CIFAR-100 into 10 subsets. Figure 5(a) shows the average accuracy and size of models at the end of the 10 tasks learned by firefly descent, Learn-to-Grow, CPG and other CL baselines. We can see that firefly descent learns smaller networks with higher accuracy. To further compare with CPG, we follow the setting of their original paper (Hung et al., 2019a) and randomly partition CIFAR-100 to 20 subsets of 5 classes to construct 20 tasks. Table 2 shows the average accuracy and size learned at the end of 20 tasks. Extra growing epochs refers to the epochs used for selecting the neurons for the next upcoming tasks, and `Individual` refers to training a different model for each task. We can see that

firefly descent learns the smallest network that achieves the best performance among all methods. Moreover, it is more computationally efficient than CPG when growing and picking the neurons for the new tasks. Figure 5(b) shows the average accuracy over seen tasks on the fly. Again, firefly descent outperforms CPG by a significant margin.

| Method | Param (M) | Extra Growing Epochs | Avg. Accuracy (20 tasks) |
|---|---|---|---|
| Individual | 2565 | - | 88.85 |
| CPG | 289 | 420 | 90.75 |
| CPG w/o FC [5] | 28 | 420 | 90.58 |
| Firefly | **26** | **80** | **91.03** |

Table 2: 20-way split continual image classification on CIFAR-100.

## 4 Related Works

In this section, we briefly review previous works that grow neural networks in a general purpose and then discuss existing works that apply network growing to tackle continual learning.

**Growing for general purpose** Previous works have investigated ways of knowledge transfer by expanding the network architecture. One of the approaches, called Net2Net (Wei et al., 2016), provides growing operations for widening and deepening the network with the same output. So whenever the network is applied to learn a new task, it will be initialized as a functional equivalent but larger network for more learning capacity. Network Morphism (Wei et al., 2016) extends the Net2Net to a broader concept, which defines more operations that change a network's architecture but maintains its functional representation. Although the growing methods are similar to ours, in these works, they randomly or adopt simple heuristic to select which neurons to grow and in what direction. As a result, they failed to guarantee that the growing procedure can finally reach a better architecture every time. (Elsken et al., 2017) solve this problem by growing several neighboring networks and choose the best one after some training and evaluation on them. However, this requires comparing multiple candidate networks simultaneously.

On the other hand, recently, (Liu et al., 2019) introduces the Splitting Steepest Descent, the first principled approach that determines which neurons to split and to where. By forming the splitting procedure into an optimization problem, the method finds the eigen direction of a local second-order approximation as the optimal splitting direction. However, the method is restricted to only splitting neurons. Generalizing it to special network structure requires case-by-case derivation and it is in general hard to directly apply it on other ways of growing. Moreover, since the method evaluates the second-order information at each splitting step, it is both time and space inefficient.

**Growing for continual learning** continual learning is a natural downstream application of growing neural networks. ProgressiveNet (Rusu et al., 2016) was one of the earliest to expand the neural network for learning new tasks while fixing the weights learned from previous tasks to avoid forgetting. LwF (Li & Hoiem, 2017) divides the network into the shared and the task-specific parts, where the latter keeps branching for new tasks. Dynamic-expansion Net (Yoon et al., 2017) further applies sparse regularization to make each expansion compact. Along this direction, Hung et al. (2019b,a) adopt pruning methods to better ensure the compactness of the grown model. All of these works use heuristics to expand the networks. By contrast, `Firefly` is developed as a more principled growing approach. We believe future works can build theoretical analysis on top of the `Firefly` framework.

## 5 Conclusion

In this work, we present a simple but highly flexible framework for progressively growing neural networks in a principled steepest descent fashion. Our framework allows us to incorporate various mechanisms for growing networks (both in width and depth). Furthermore, we demonstrate the effectiveness of our method on both growing networks on both single tasks and continual learning problems, in which our method consistently achieves the best results. Future work can investigate various other growing methods for specific applications under the general framework.

## Broader Impact

This work develops a new framework that can grow neural networks simply and efficiently, which can be generally used in various applications that using neural networks and positively enhance their capacity and performance. In particular, we anticipate it can be applied on devices which have hard memory/computation constraint, i.e. mobile devices or robots. Our work does not have any negative societal impacts.

## Funding Transparency Statement

**Related Funding:** NSF (CPS-1739964, IIS-1724157, NRI-1925082, CAREER-1846421, SenSE-2037267, EAGER-2041327), ONR (N00014-18-2243), FLI (RFP2-000), ARO (W911NF-19-2-0333), DARPA, Lockheed Martin, GM, and Bosch.

Peter Stone serves as the Executive Director of Sony AI America and receives financial compensation for this work. The terms of this arrangement have been reviewed and approved by the University of Texas at Austin in accordance with its policy on objectivity in research.

## Footnotes

[2] A neuron is determined by both $\sigma$ and $\theta$. But since $\sigma$ is fixed under our discussion, we abuse the notation and use $\theta$ to represent a neuron.

[3]Because all we need to store is the gradient, which is of the same size as the original parameters.

[4]It is also possible to introduce new layers for continual learning, which we leave as an interesting direction for future work.

[4]CPG without fully connected layers is to align the model structure and model size with `Firefly`.

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
