[Supplementary Material]

# A    Detailed Algorithm for Continual Learning

Algorithm 2 summarizes the pipeline of applying firefly descent on growing neural architectures for continual learning problems.

---

**Algorithm 2** Firefly Steepest Descent for Continual Learning

---

    **Input** : A stream of datasets $\{\mathcal{D}_1, \mathcal{D}_2, \ldots, \mathcal{D}_T\}$;
    **for** task $t = 1 : T$ **do**
      **if** $t = 1$ **then**
        Train $f_1$ on $\mathcal{D}_1$ for several epochs until convergence.
        Set mask $m_1$ to all 1 vector over $f_1$.
      **else**
        Denote $f_t \leftarrow f_{1:t-1}$ and lock its weights.
        Train a binary mask $m_t$ over $f_t$ on $\mathcal{D}_t$ for several epochs until convergence.
      **end if**
      $f_t = f_t[m_t]$   // $f_t$ is re-initialized as the selected old neurons from $f_{1:t-1}$ with their weights fixed.
      **while** $f_t$ can not solve task $t$ sufficiently well **do**
        **if** $t = 1$ **then**
          Grow $f_t$ by **splitting** existing neurons and growing new neurons.
        **else**
          Grow $f_t$ by **unlocking** existing neurons and growing new neurons.
        **end if**
        Train $f_t$ on $\mathcal{D}_t$
      **end while**
      Update $m_t$ as the binary mask over $f_t$.
      Record the network mask $m_t$, $f_{1:t} = f_{1:1-t} \cup f_t$.
    **end for**

---

# B    Experiment Detail

## B.1    Toy RBF Network

We construct a following one-dimensional two-layer radial-basis function (RBF) neural network with one-dimensional inputs,

$$f(x) = \sum_{i=1}^{m} w_i \sigma(\theta_{i1} x + \theta_{i2}), \quad \text{where} \quad \sigma(t) = \exp\left(-\frac{t^2}{2}\right), \quad x \in \mathbb{R}, \tag{6}$$

where $w_i \in \mathbb{R}$ and $\theta_i = [\theta_{1i}, \theta_{2i}]$ are the input and output weights of the $i$-th neuron, respectively. We generate our true function by drawing $m = 15$ neurons with $w_i$ and $\theta_i$ i.i.d. from $\mathcal{N}(0, 3)$. For dataset $\{x^{(\ell)}, y^{(\ell)}\}_{\ell=1}^{1000}$, we generate them with $x^{(\ell)}$ drawing from $\text{Uniform}([-5, 5])$ and let $y^{(\ell)} = f(x^{(\ell)})$. We apply various growing methods to grow the network from one single neuron all the way up to 12 neurons.

For the new initialized neurons introduce during the growing in `RandSearch` and `Firefly`, we draw the neruons from $N(0, 0.1)$. For `RandSearch`, we finetune all the randomly grow networks for 100 iterations. For `Firefly`, we also train the expanded network for 100 iterations before calculating the score and picking the neurons. Further, We update 10,000 iterations between two consecutive growing.

## B.2    Growing Wider and Deeper Networks

**Setting for Growing Wider Networks**    For all the experiment including Net2Net, splitting steepest descent, NASH and our firefly descent, we grow 30% more neurons each time. Between two consecutive grows, we finetune the network for 160 epochs.

For splitting steepest descent, we follow exactly the same setting as in Liu et al. (2019).

For NASH, we only apply "Network morphism Type II" operation described in Elsken et al. (2017), which is equivalent to growing the network width by randomly splitting the existing neurons.. During the search phase, we follow the original paper's setting, sample 8 neighbour networks, train each of them for 17 epochs and choose the best one as the grow result.

For firefly descent, we grow a network by both splitting existing neurons and adding brand new neurons for widening the network; When growing, we split all the existing neurons and add $m' = 50$ brand new neurons draw from $N(0, 0.1)$. We will also train the expanded network for 1 epoch before calculating the score and picking the neurons.

**Growing Wider MobileNet V1**    We also compare firefly with other growing method on MobileNet V1 using CIFAR-100 dataset. Same as Wu et al. (2020), we start from a thinner MobilNet V1 with 32 channels in each layer. We grow 35% more neurons each time, the other settings are same as the previous growing wider networks' setting.

Figure 6: Results and time consumption of growing increasingly wider networks on CIFAR-100 using MobileNet V1 backbone

Figure 6 again shows that firefly splitting can out perform various of growing baseline on the same backbone network. Meanwhile, its time cost is much smaller than splitting and NASH algorithm.

**Growing Deeper Networks**    We test firefly descent for growing network depth. We build a network with 4 blocks. Each block contains numbers of convolution layers with kernel size 3. The first convolution layer in each block is stride two. For a simple and clear explanation, we mark the number of layers in these 4 blocks as 12-12-12-12, for example, which means each block contains 12 layers. Begin from 1-1-1-1, we grow the network using firefly descent on MNIST, FashionMNIST, SVHN, and compare it with AutoGrow Wen et al. (2019) and NASH Elsken et al. (2017).

For our method, we start from a 1-1-1-1 network with 16 channels in each layer. We also insert 11 identity layers in each block, which roughly match the final number of layers in AutoGrow. We apply our growing layer strategy described in Section 2.3 for growing new layers and apply both splitting existing neurons and adding brand new neurons for widening the existing layers. When growing new layers, we introduce $m' = 20$ new neurons in each Identity map layers, when increasing the width of the existing layers, we split all the existing neurons and add $m' = 20$ new neurons. After expanding the network, we train the network for 1 epoch before calculating the score. If the Identity layer remains 2 or more new neurons after selection, we add this Identity layers in the network and train with the existing network together. Otherwise, we will remove all the new neurons and keep this layer as an Identity map. For the existing neurons, we grow 25% of the total width.

For NASH, we apply "Network morphism Type I" and "Network morphism Type II" together, which represent growing depth by randomly insert identity layer and growing width by randomly splitting the existing neurons. During the search phase, we follow the original paper's setting, sample 8 neighbor networks, train each of them for 17 epochs and choose the best one as the growing result. Each time when sampling the neighbour networks, we grow the total width of the existing layers by 25% and then randomly insert one layer in each blocks.

For both our method and NASH, we grow 11 steps and finetune 40 epochs after each grow step. We also retrain the searched network for 200 epochs after the last grow to get the final performance on each dataset.

For AutoGrow, we use the result report in the original paper.

Table B.2 shows the result. We can see our method can grow a smaller network to achieve the AutoGrow's performance and outperform the network searched with NASH.

| Dataset | Method | Structure | Param (M) | Accuracy |
|---|---|---|---|---|
| | AutoGrow Wen et al. (2019) | 13-12-12-12 | 2.3 | 99.57 |
| MNIST | NASH Elsken et al. (2017) | 12-12-12-12 | 2.0 | 99.50 |
| | Firefly | 12-12-12-12 | **1.9** | **99.59** |
| | AutoGrow Wen et al. (2019) | 13-13-13-13 | 2.3 | 94.47 |
| FashionMNIST | NASH Elsken et al. (2017) | 12-12-12-12 | 2.2 | 94.34 |
| | Firefly | 12-12-12-12 | **2.1** | **94.48** |
| | AutoGrow Wen et al. (2019) | 12-12-12-11 | 2.2 | 97.08 |
| SVHN | NASH Elsken et al. (2017) | 12-12-12-12 | 2.0 | 96.90 |
| | Firefly | 12-12-12-12 | **1.9** | 97.08 |

Table 3: Result on growing Depth comparing with two baselines

## B.3 Application on Neural Architecture Search

Following the setting in DARTS (Liu et al., 2018b), we separate half of the CIFAR-10 training set as the validation set for growing. We start with a stacked 5 cell network for searching, the second and the fourth cell are reduction cells, which means all the operations next to the input of the cells are set to stride two. In each cell, we build the SepConv and DilConv operation blocks following DARTS (Liu et al., 2018b). To apply our firefly descent, we grow the last convolution layer in each block and add a linear transform layer with the same output channels to ensure all the operations on the same edge can sum up in the same size as the output. The number of channels of the operations in each cell is set to 4-8-8-16-16, which is $0.25\times$ of that in the original Darts. The last linear transform layer in each cell has channels 16-32-32-64-64. We grow the network by both splitting existing neurons and adding brand new neurons, and each time we sequentially select one cell to grow. We repeat growing the whole 5 cells twice, which means we apply our firefly descent for 10 times in total. Each time, we split all the existing neurons in the chosen cell and add 4, 8, 8, 16, 16 brand new neurons differently for the 5 cells. We then train the expanded network for 5 epochs and select $25\%$ neurons to grow. As a result, we search the network structure for 100 epochs in total. All other training hyperparameters are set to the same values as in DARTS (Liu et al., 2018b).

After searching, we select the operation with the largest width in each edge as the final operation. Besides, if all operations on the same edge grow less than $20\%$ comparing to the initial width, we assign this edge as Identity map in the final structure. We only keep the type of operations in the cell as our final search result because we need to increase the channel width to match the model size with the baselines.

For the final evaluation, we sequentially stack a 20 cell network and mark those cells as 1-20. We apply the search result of the first, second, third, fourth, and the fifth cell in the 5 stacked search network to cell 1-6, cell 7, cell 8-13, cell 14, and cell 15-20 of the final evaluation network accordingly. We increase the initial channel to 40 to match the model size with other baselines. The other training settings are kept the same as in DARTS (Liu et al., 2018b). Our result is averaged over 5 runs from our final evaluation model.

## B.4 Continual Learning

For both 10-way split CIFAR-100 and 20-way split CIFAR-100, we repeat the experiment 3 times with 3 different task splits. We apply both the copy-exist-neuron and grow-new-neuron strategies to tackle the CL problem. During each growing iteration, we add 15 brand new neurons for each layer as candidates for growing. After expanding the network, we finetune the network for 50 epochs on the new task. During the selection phase, for 20-way split CIFAR-100, we select out the top 256 neurons among all the copied neurons and new neurons. For 10-way split CIFAR-100, we select the top $32, 128, 196, 256, 320, 384, 448, 512$ neurons each time to test our performance under different model size. After selecting the neurons, we finetune the expanded network on the new task for 100 epochs.