[Reviews · NeurIPS 2020]

Review 1

Summary and Contributions: This paper propose an iterative optimization solution to grow neural networks in the architecture neighborhood to jointly optimize the network’s parameters and architecture. The proposed framework achieves similar or better performance compared to related works. The main contribution of this paper is defining an architecture search neighborhood and extending the searching by splitting existing neurons to jointly optimize splitting existing neurons, growing new neurons and growing new layers, and optimize the network parameter sets and architecture iteratively.

Strengths: +This paper introduces a method to apply “architecture descent” to iteratively search the network parameter set and architecture neighborhood to find the next optimal architecture; the theoretical grounding is sound. +This paper extends the network growing method from splitting existing neurons to include search of growing new neurons and growing new layers. It proposes a novel approach to find a functional neighborhood. +The baseline comparison experiments show the performance comparable to state of arts with similar accuracy and better search time.

Weaknesses: -This paper did not compare directly to the most related growing networks [1]. Instead it compared with 2 years ago's work (DARTS). -This paper only validate experiments on CIFAR-10 and CIFAR-100 which cannot explain the performance of the found architecture. - If the paper didn't validate on Imagenet, at least it can be validated on NAS-Bench-101, 201 and others. [2,3]. But this paper didn't provide a comprehensive study of these comparison. - Therefore, this experiments can't support the claim of "learns networks that are smaller in size but have higher average accuracy than those learned by the state-of-the-art methods." [1] AutoGrow: Automatic Layer Growing in Deep Convolutional Networks, KDD2020 [2] NAS EVALUATION IS FRUSTRATINGLY HARD, ICLR2020 [3]Ying, Chris, et al. "Nas-bench-101: Towards reproducible neural architecture search." International Conference on Machine Learning. 2019.

Correctness: Yes

Clarity: Yes

Relation to Prior Work: Yes

Reproducibility: Yes

Additional Feedback:


Review 2

Summary and Contributions: ################################################ I have read the rebuttal and other reviews. I think the authors adequately addressed these points in the response, and I would lean towards acceptance at this point. ################################################ The paper proposes an algorithm for growing neural networks while jointly optimizing the network parameters and architecture. The architectures can be grown more flexibly and to be more resource-efficient than prior work.

Strengths: (1) The paper is well written. (2) Optimizing neural network architectures is an important problem (3) Dynamically growing neural networks online is an important problem in optimizing network architecture. (4) The algorithm seemed to provide a reasonable performance boost for continual learning.

Weaknesses: (1) It’s unclear to me if the results in “Cell-Based Neural Architecture Search” are significant. ENAS/DARTS frequently perform worse than random architectures from their search space - [1]. But, these are also fairly different architecture search algorithms. Thus, I believe the key number is the difference between the random search. However, I would want more details about the random search is performed to see if this is reasonable. (2) How are the hyperparameters for your algorithm selected? (3) It would be nice to have an explicit related work section. [1] Adam, George, and Jonathan Lorraine. "Understanding neural architecture search techniques." arXiv preprint arXiv:1904.00438 (2019).

Correctness: The claims and methods seem correct to me.

Clarity: The paper is well written and clear for the most part - I think this is a strength of the paper.

Relation to Prior Work: It is difficult to assess how the work differs from previous contributions. The differences are mixed throughout the paper, but I would have appreciated a related work section.

Reproducibility: Yes

Additional Feedback: These comments did not affect my score but may help improve the paper. 29: growing -> growth 213: splittting -> splitting


Review 3

Summary and Contributions: The paper proposes a growing algorithm (Firefly) to increase the width and depth of a neural network. The Firefly increases the width/depth by using the idea of Splitting Steepest Descent [10] and adding brand new neurons/layers. The effectiveness of the method is demonstrated in a broad scope of applications such as Neural Architecture Search, Continual Learning, and searching efficient and accurate models, comparing with state-of-the-art work. The two key factors of the success are: (1) a gradient-based optimization, which optimally identifies which new neurons/layers can minimize the loss more; (2) newly added neurons/layers drop in gradually near the local minimum of the smaller net without dramatically change the loss of the small net.

Strengths: 1. a well explainable growing algorithm with efficient growing speed; 2. experiments on many applications comparing with state-of-the-art work.

Weaknesses: The writing quality and missing details degrade my rating of the paper: 1. The mismatch between math equations and intentions: 1.1. In Line 90, I bet f_t (x) should be the final output (a vector/scalar) of a net, but the sum is just an input to a neuron in the next layer; 1.2. The use of \epsilon and \varepsilon is ambiguous. I bet \epsilon is a small threshold such that new neurons don't change the loss much. However, in Line 101 & 105, I bet \varepsilon should be used. 1.3. in Step Two between Line 121 and Line 122, for a standard Taylor approximation, I bet s_i should be just a \Delta L. Please explain why it is an integration. 1.4. between Line 166 and Line 167, readers can be confused if f_{1:t} (x) is a sequence of functions from step 1 to step t, or just a function at step t. If the latter, what's its difference from f_t (x)? 1.5. "the candidate set of f_t+1 should consist of" is confusing. Why a set is just a function? 2. some important but missing details (see comments in the "Clarity")

Correctness: No crucial errors.

Clarity: 1. clarify if \varepsilon and \delta are learnable parameters the same as model parameters or they are just learnable during the architecture descent. In Line 116, when optimizing \varepsilon and \delta, are neural network weights also updated? 2. "measured by the gradient magnitude", magnitude of full-batch or a few mini-batches? 3. clarify if L is training loss or validation loss. 4. clarify z in Line 124. A small number? 5. Make use the legend labels "Random (split)" and "RandSearFh" in Fig. 3(a) are exactly the same with those appeared in the text ("RandSearch (split)" and "RandSearch (split+new)"). In Fig. 3(a), a should-have simple baseline: add one neuron and randomly initialize new weights. 6. In Figure 3(b), If the splitting and growing happen at the same time, the number of neurons (markers along x-axis) should have a gap larger than 1. Why did the markers appear at all x locations? 7. In Line 207, clarify the depth of the net. 8. In Figure 5/Line 249, cite and clarify baselines of EWC, DEN and RCL, where are not clarified/mentioned anywhere. Moreover, why other baselines are not curves but single dots. 9. The x-axis with 20 tasks doesn't match the caption "on 10-way split"

Relation to Prior Work: Introduction includes some related works. Related Work can be added if there is space and you might. Some content between Line 166 and 170 is basically a repeat. Trimming this can release some space. Consider the following related work: Dai, Xiaoliang, Hongxu Yin, and Niraj K. Jha. "NeST: A neural network synthesis tool based on a grow-and-prune paradigm." IEEE Transactions on Computers 68, no. 10 (2019): 1487-1497. Philipp, George, and Jaime G. Carbonell. "Nonparametric neural networks." arXiv preprint arXiv:1712.05440 (2017).

Reproducibility: No

Additional Feedback: 1. Line 97, \theda should be \theda_i 2. “A neurons is determined” -> "A neuron is determined" ================== Thanks for the clarification. Recap: this is an interesting paper with a broad of experimental evaluations. If the AC trusts the authors will improve the writing quality with (lots of) efforts, my final rate will go to 7.


Review 4

Summary and Contributions: This paper proposed a simple but highly flexible framework for progressively growing neural networks in a principled steepest descent fashion. The paper also demonstrates the effectiveness of our method on both growing the network on a single task and continual learning problems.

Strengths: The paper is easy to understand and interesting. The empirical evaluation is also quite convincing. Neural network architecture search is an important topic.

Weaknesses: There is no discussion of time complexity or space complexity of the proposed method. The method seems quite complicated and difficult to implement. And it's also surprising that the method can achieve better performance with shorter time usage.

Correctness: Yes

Clarity: Yes

Relation to Prior Work: The author discussed previous works and also have strong baselines.

Reproducibility: Yes

Additional Feedback:

[Author Response · NeurIPS 2020]

We sincerely thank all reviewers for their valuable comments and we address individual questions below.

**R1: This paper only compares to the 2 years ago's work (DARTS).** We actually compare with AutoGrow in
paragraph **Growing Wider and Deeper Networks**, around line 218. Due to the space limit, we move the re-
sult to appendix. Moreover, AutoGrow can only grow layers so it does not compare with many well known
NAS baseline. We choose DARTS because it is still a very popular baseline in recent NAS papers, which
gives new papers a fast, gradient based, weight-sharing NAS baseline with a low error rate to compare with.

**R1: The paper only experiments on CIFAR-10/100**
**but not on Imagenet. At least it can be validated on**
**NAS-Bench-101,201.** Thanks for the suggestion, we
have compared the firefly method with similar weight
sharing baseline methods on NAS-Bench-201 in Table 1.

| | RSPS | DARTSV1/2 | ENAS | SETN | GDAS | Ours |
|---|---|---|---|---|---|---|
| Acc. | 84.07 | 54.30 | 53.89 | 87.64 | 93.61 | 93.27 |

**Table 1:** Search Result on NAS-Bench-201

We also want to point out that firefly achieves very good results on continual learning (CL), outperforming the best
known dynamic architecture baselines (CPG, DEN) in the CL literature.

**R2: ENAS/DARTS are fairly different architecture search algorithms. I want more**
**details on how random search is performed.** Thanks. In the paper (Table 1 line 240),
we reported the random search results from the DARTS paper. For a more comprehensive
comparison, we add a detailed random search experiment here in Figure 1 which searches
different numbers of random samples and evaluates the best one on the validation set.

**Figure 1:** Err. v.s. search time (in GPU days).

**R2: hyperparameters?** Due to the complexity of the NAS problem, certain number of
hyperparameters is unavoidable. Our method is already much simpler than alternatives such
as RL-based methods and DARTS. We mainly have three key hyperparameters: how many new neurons we add, how
many steps we train after adding these new neurons and how many candidate neurons we choose after training. We
provided the exact numbers we used for each experiment in Appendix C.

**R2: It would be nice to have an explicit related work section.** We will move the related work to main content.

**R3: Notations: 1) line 90 $f_t(x)$ should be the final output (a vector/scalar) of a net, but the sum is just an input**
**to a neuron in the next layer.** $f_t$ here is a simple two-layer network that takes $x$ and outputs a scalar; we followed
the similar definition as in reference [10] in the original paper. **2) the use of $\epsilon$ and $\varepsilon$ is ambiguous.** Thanks, we will
make the usage consistent. **3) in Step Two (line 121-122), for a standard Taylor approximation, $s_i$ should be just**
**a $\Delta$.** We used a different version of Taylor expansion here; note that our $s_i$ is close to gradient $\nabla_{\xi_i} L(f_{[\tilde{\varepsilon}_{-i},0]}, \tilde{\delta}))$ when
$\tilde{\varepsilon}_i$ is close to zero. **4) line 166-167, is $f_{1:t}(x)$ is a sequence of functions or just a function at step $t$?** $f_{1:t}$ is a single
network at step $t$ that combines all neurons grown from step 1 to $t$, and $f_t$ is the subnetwork of $f_{1:t}$ selected by a binary
mask only for task $t$. **5) "the candidate set of $f_{t+1}$ should consist of" is confusing. Why a set is just a function?**
Thanks, we will make it explicitly refer to the set of functions indexed by parameters $\varepsilon, \delta$.

**R3: Clarifications:** We will improve the clarity based on your suggestions. **1) line 116, when optimizing $\varepsilon$ and**
**$\delta$, are neural network weights also updated?** No, the network $\theta$ is fixed. We are only learning the perturbation
$\varepsilon\delta$. **2) "measured by the gradient magnitude", magnitude of full-batch or a few mini-batches?** We use a few
mini-batches for estimation. **3) clarify if $L$ is training loss or validation loss.** $L$ is the training loss throughout
the paper. **4) clarify $z$ in Line 124.** $z$ is a dummy variable ranging from 1 to $n$; here $\{(2z-1)/2n\}_{1 \le z \le n}$ is the
discretization of the continuous range $[0,1]$. **5) Make the legend labels the same with those appeared in the text. In**
**Fig. 3(a), a should-have simple baseline: add one neuron and randomly initialize new weights** Thanks, we will
make them consistent. The suggested baseline is strictly worse than the Random(split) but we will include that as
well. **6) In Figure 3(b), If the splitting and growing happen at the same time, the number of neurons (markers**
**along x-axis) should have a gap larger than 1** We pick the single best neuron from splitting and growing, so they
do not happen simultaneously. **7) In Line 207, clarify the depth of the net** For VGG-19, the depth is fixed to be 19.
**8) In Figure 5/Line 249, cite and clarify baselines of EWC, DEN and RCL, where are not clarified/mentioned**
**anywhere. Moreover, why other baselines are not curves but single dots** We will include the citations. We reported
the numbers for the architectures from the original papers so they are only dots. **9) The x-axis with 20 tasks doesn't**
**match the caption "on 10-way split"** We have corrected the typo.

**R4: No discussion of time/space complexity.** We discussed the space complexity on line 150. The time complexity
per expansion is $\mathcal{O}(N+m)$, where $N$ is the size of the sub-network we consider expanding and $m$ is the number of
new neuron candidates. Because per expansion, we only compute the gradient for each neuron, thus the complexity is
linear. We will add more detailed discussion in the revision.

**R4: The method seems complicated and difficult to implement** It is in fact easier to implement Firefly than other
NAS approaches because 1) firefly only requires gradient estimation, for which standard deep learning libraries have
APIs; 2) the search space is smaller than conventional NAS methods because every expansion is restricted to be local.
The core idea of Firefly is to train the local perturbations of a given architecture which leads to the steepest descent.
The entire implementation of both growing/splitting is only a few hundred lines of python code. We have built our
method in an API fashion and will open source the code.

[Meta-Review · NeurIPS 2020]

Looking over the reviews, rebuttal and discussion afterwards, I think this paper is of interest to the community. However there are several ambiguities and issues in the presentation of the work that had been noted by the reviewers (e.g. R3) which does hamper how well the work can be understood. So I *urge* the authors to make sure all the points raised by the reviewers are considered (as mentioned in the rebuttal) and incorporated in the main text and make sure the text is clear and easy to parse. I do believe the rebuttal addresses most of the issues brought up, in particular I think the replies given to the issues raised by R1 are valid, including additional experiments that were required by the reviewer. So I think overall the positive outweighs the negatives. But do incorporate the rebuttal in the paper, please do take care of everything that was raised in the reviews. They will only make the paper stronger, and make sure it will have the impact it deserves.